# The Effect of γ-Aminobutyric Acid Addition on In Vitro Ruminal Fermentation Characteristics and Methane Production of Diets Differing in Forage-to-Concentrate Ratio

**Yan-Lu Wang, Zhi-Hui Zhang, Wei-Kang Wang** [ID]**, Qi-Chao Wu, Fan Zhang, Wen-Juan Li, Sheng-Li Li, Wei Wang, Zhi-Jun Cao** [ID] **and Hong-Jian Yang \*** [ID]

State Key Laboratory of Animal Nutrition, College of Animal Science and Technology,
China Agricultural University, Beijing 100193, China
\* Correspondence: yang_hongjian@sina.com; Tel.: +86-13911888062 or +86-139118880622

**Abstract:** Gamma-aminobutyric acid (GABA), known as the most abundant inhibitory neurotransmitter in the mammalian brain, can permeate ruminal epithelia by passive diffusion and enrich in the rumen environment. To explore whether the addition of GABA can regulate rumen fermentation characteristics as well as methane production, a 2 × 6 factorial in vitro rumen batch culture was conducted to determine the supplemental effect of GABA at inclusion levels of 0 (Control), 10, 20, 30, 40 and 50 mg in culture fluids on rumen fermentation of two total mixed rations (HF—a high-fiber ration consisted of 70% corn silage and 30% concentrate; and LF—a low-fiber ration consisted of 30% corn silage and 70% concentrate). After 72 h in vitro incubation of two rations with mixed rumen microorganisms obtained from five rumen-cannulated lactating Holstein dairy cows, increasing GABA addition linearly increased cumulative gas production in the LF group, though in vitro dry matter digestibility was not affected in either the LF or HF group. Kinetic gas production analysis noted that increasing GABA addition mostly decreased the gas production rate (i.e., RmaxG), as well as the ration digestion rate (RmaxS) to reach maximum fermentation. The GABA addition did not affect pH or microbial growth (i.e., MCP). However, total volatile fatty acid production in both LF and HF groups all linearly increased with the increase in GABA addition. Along with the increase in GABA addition in both LF and HF groups, the ratio of non-glucogenic to glucogenic volatile fatty acids both increased, while the molar proportions of propionate and valerate were significantly decreased, and the acetate and butyrate proportions were increased after 72 h in vitro rumen fermentation. The time-course change of fermentation end-products generally showed that carbon dioxide declined from approximately 89% to 74%, and methane increased from approximately 11% to 26%. After 72 h in vitro fermentation, molar methane proportion was greater in the LF than in the HF group, and increasing GABA addition quadratically increased methane production in the LF group while a slight increase occurred in the HF group.

**Keywords:** gamma-aminobutyric acid; in vitro rumen fermentation; gas production

## 1. Introduction

From the basic science viewpoint, "microbial endocrinology" deals with those theories in which neurochemicals, produced by both multicellular organisms and prokaryotes, are considered as a common shared language that enables interkingdom communication. Gut microbiota provides the host with multiple functions and interacts with the host organism through both direct contact and neuroactive molecules, which are produced during microbial metabolism. Gamma-aminobutyric acid (GABA) is a four-carbon non-protein amino acid (Figure 1), and is known as the most abundant inhibitory neurotransmitter in the mammalian brain [1,2]. It can be produced by the enzymatic decarboxylation of glutamate which has been found not only in mammals but also in insects, bacteria and, plants [3].

Several lactic acid bacteria such as *Escherichia coli*, *Pseudomonas*, and *Bifidobacterium* strains have also been reported to biosynthesize GABA [4–7].

**Figure 1.** Molecular structure of gamma-aminobutyric acid (GABA).

Rackwitz and Gäbel noted that GABA was able to permeate ovine ruminal and jejunal epithelia of sheep, mainly by passive diffusion mounted in Ussing chambers [8]. The addition of GABA has shown some beneficial effects such as improving growth performance, feed intake, milk performance, reducing disease occurrence, and alleviating heat stress on ruminants [9–12]. In the last several decades, GABA has increasingly been applied as a feed additive for ruminants either to augment performance or to reduce the side effects of high-energy diets in the majority of the previous aforementioned studies [9–12]. In recent years, however, the supplementation of GABA came to be controversial. For instance, Matsumoto et al. reported that the addition of GABA preparation could improve the general health condition and growth rate of suckling Japanese Black beef calves [9]. Conversely, Dawson and Mayne reported that the infusion of GABA in steers did not affect dry matter intake and feed behavior, and no beneficial effect was observed [13].

It is well known that the rumen harbors populations of the three domains of Bacteria, Archaea (i.e., methanogens) and Eucarya (i.e., protozoa and fungi), and they play important roles in the nutrient digestion of whatever high- or low-fiber ration is fed to the host ruminants. For many years, nutritionists, microbiologists, and physiologists have studied the rumen with the aim being to maximize productivity and improve overall host health by manipulating the rumen and its microbial ecosystem. In the above limited studies, the use of GABA mainly focused on its response in terms of health, growing performance, feed intake, and feeding behavior in beef calves or steers. However, it is unclear until now whether or not GABA addition could regulate fermentation behavior of rumen microorganisms in terms of nutrient digestion, rumen production of volatile fatty acids (VFA), and methane production.

An in vivo animal trial study of the rumen involving VFA absorption through ruminal epithelia and eructation of fermentation end-gases has been conducted to mimic rations differing in forage-to-concentrate ratio practically applied in feeding ruminants [14]. Considering this, an in vitro rumen batch culture in terms of kinetic gas production, VFAs and methane production was applied in the present study. The primary objective was to elucidate whether or not the addition of GABA at different dosages could alter rumen fermentation characteristics, and hopefully such a study could provide us scientific insights to better understand the potential action mode of GABA in rumen fermentation.

## 2. Materials and Methods

### 2.1. Animals, Experimental Rations and Feed analysis

Five rumen-cannulated lactating Holstein dairy cows (4 years old, $60 \pm 13$ days in milk, body weight of $543 \pm 45$ kg, and daily yield of $18.47 \pm 0.77$ kg) were housed in a free stall and served as rumen fluid donors. The cows had free access to drinking water, and were daily fed 20 kg corn silage, 2.5 kg Chinese wildrye hay, and 8.0 kg commercial concentrate consisting of 530 g/kg corn meal, 140 g/kg soybean meal, 70 g/kg cottonseed meal, 40 g/kg rapeseed meal, 120 g/kg distillers dried grains, 10 g/kg limestone, 10 g/kg dicalcium phosphate, 10 g/kg NaCl, 10 g/kg sodium bicarbonate, and 10 g/kg trace mineral and 50 g/kg vitamin premix.

As shown in Table 1, to prepare two experimental diets for subsequent in vitro fermentation, whole-corn silage and the commercial concentrate for lactating dairy cows

were chosen as forage and concentrate, respectively. The rations were mixed with forage-to-concentrate ratios of 70:30 (HF, a high-fiber ration) and 30:70 (LF, a low-fiber ration). Representative samples of HF and LF were dried in a fan-assisted oven (DHG-9420A, YiHeng Scientific Instrument Limited Company, Shanghai, China) at 65 °C for 48 h, and then ground to pass through a 1 mm screen. The crude protein (N × 6.25), ether extract, and ash contents in diet samples were analyzed following the procedures of the Association of Official Analytical Chemists [15]. Neutral detergent fiber and acid detergent fiber content were determined according to the method of Van Soest et al. [16].

**Table 1.** Feed ingredients and chemical composition on dry matter basis of high-fiber (HF) and low-fiber (LF) diets for in vitro rumen fermentation.

| Item | HF | LF |
|---|---|---|
| Ingredient | | |
| Whole-corn silage | 700 | 300 |
| Concentrate [1] | 300 | 700 |
| Nutrient level (g/kg DM) | | |
| Crude protein | 116.9 | 175.8 |
| Neutral detergent fiber | 425.4 | 244.0 |
| Acid detergent fiber | 25.58 | 158.0 |
| Ether extract | 15.2 | 13.4 |
| Ash | 57.9 | 69.7 |

[1] The supplemental concentrate per kg consisted of 530 g corn meal, 140 g soybean meal, 70 g cottonseed meal, 50 g wheat bran, 40 g rapeseed meal, 120 g corn distiller's dried grains with solubles, 10 g limestone, 10 g dicalcium phosphate, 10 g sodium bicarbonate, 10 g sodium chloride, and 10 g vitamins and trace minerals.

### 2.2. In Vitro Rumen Fermentation

#### 2.2.1. Experimental Design

A 2 × 6 factorial experimental design was applied to in vitro batch cultures with the 2 diets (HF, LF) and six GABA addition levels in each diet group at 0 (control), 10, 20, 30, 40, 50 mg in culture fluid, resulting in 0, 20, 40, 60, 80, 100 mg/g of dietary DM substrate, respectively. Each dosage was arranged in five fermentations, and the incubation was repeated in three experimental runs. The GABA was purchased from Sigma (Ronkonkoma, NY, USA). All chemicals in the present study were purchased from the Beijing Chemical Reagent Company (Beijing, China).

#### 2.2.2. Rumen Fluid Collection

Rumen fluids from each cow were collected through the fistulas 1 h before morning feeding from different sites inside the rumen, squeezed through 4 layers of medical-use cheesecloth and mixed in equal proportion. The rumen fluid was transferred into a thermos pre-warmed at 39 °C and served as the inoculant for further in vitro rumen fermentation. All the animals were cared for and experimental operations were carried out under the Guidelines of Animal Care Committee and animal welfare guidelines of China Agricultural University (CAU20171014-1).

#### 2.2.3. In Vitro Rumen Fermentation and Sampling

The in vitro digestibility of DM (IVDMD) and kinetic gas production were performed according to Pang's method [17]. The specific steps for each run of the incubation were done as follows: The medium (pH 6.85) was prepared using Menke's method and pre-warmed at 39 °C in a water bath [18]. The incubators used in the present study were 120 mL glass bottles with Hungate's stoppers and screw caps. According to the experimental design, 500 mg of diet substrate, 50 mL of pre-warmed medium, and 25 mL of strained rumen fluids were mixed into each incubator (2 TMR × 5 GABA addition levels × 5 bottles). GABA was added at 0 (control), 10, 20, 30, 40, and 50 mg in the culture fluid. Another four bottles without substrates or GABA served as blanks. All bottles were purged using $N_2$ to remove air and sealed with Hungate's stoppers and screw caps immediately. These bottles were

then immediately connected to the gas channel inlets of China-patented Automated Trace Gas Recording System (AGRS-III, China Agricultural University, Beijing, China P.R.) [19]. The fermentation gas during the 72 h incubation was collected with air bags for further gas composition analysis [20].

After 72 h rumen incubation, all the bottles were removed into cold water to stop fermentation. To detect in vitro dry matter disappearance (IVDMD), we filtered the biomass materials in every bottle with a nylon bag (8 × 12 cm, 42 μm pore size) and dried at 65 °C for 48 h. Part of the filter was used to detect the pH after 72 h rumen incubation. Other parts of the culture fluid underwent $430\times g$ centrifugation to remove protozoa and feed pellet, and were then stored at $-20$ °C for further microbial crude protein (MCP) and volatile fatty acid (VFA) analyses. The MCP concentration was measured according to Makkar's method using a microplate reader (RT-6500, Rayto Instruments, Shenzhen, China) at 595 nm wavelength [21].

The concentration of VFAs (including acetate, propionate, isobutyrate, butyrate, isovalerate and valerate) were measured using a gas chromatograph (GC522, Wufeng Instruments, Shanghai, China) equipped with a 15 m semicapillary column (⌀ 0.53 mm) packed with Chromosorb 101, with pure $N_2$ as the carrier gas, and crotonic acid as the internal standard at a column temperature of 120 °C, injector temperature of 250 °C, and a detector temperature of 250 °C. The $H_2$ flow rate was 37.5 mL/min, the air flow rate was 143 mL/min, and the carrier gas flow rate was 72 mL/min.

The concentration of $CO_2$ and $CH_4$ was measured according to Yang's method [20]. We injected 1 mL gas from the air bag into the gas chromatograph using a 2 m stainless steel column (2·0 mm inner diameter) packed with TDX-1. $N_2$ was used as the carrier gas.

### 2.3. Calculations

Cumulative gas production (GP) was fitted to a monophasic model as in Equation (1) [22]:

$$GP_t = \frac{A}{1 + \left(\frac{C}{time}\right)^B} \tag{1}$$

where $GP_t$ is the cumulative gas production (mL/g DM) at rumen fermentation time t (h), A is the asymptotic gas production (mL/g DM), B is a sharpness parameter determining the shape of the curve, and C is the time (h) at which half of the asymptotic cumulative gas volume is reached. A, B and C were calculated using the nonlinear procedure of SAS 9.4 (Cary, NC, USA).

The maximum gas production rate (RmaxG, mL/h), the time at which RmaxG is reached (TRmaxG, h), maximum rate of substrate digestion (RmaxS, /h), and the time at which the maximum rate of substrate digestion is reached (TRmaxS, h) were calculated with A–C (Yang et al., 2005) as in Equations (2)–(5):

$$RmaxG = \left(A \times C^B \times B \times TRmaxG^{(-B-1)}\right) / \left(1 + C^B \times \left(TrmaxG^{(-B)}\right)\right)^2 \tag{2}$$

$$Trmax\,G = C \times ((B-1)/(B+1))^{(1/B)} \tag{3}$$

$$RmaxS = \left(B \times TrmaxS^{(B-1)}\right) / \left(C^B + TrmaxS^B\right) \tag{4}$$

$$Trmax\,S = C \times (B-1)^{(1/B)} \tag{5}$$

The ratio of non-glucogenic to glucogenic acids (NGR) was calculated as in Equation (6) [23]:

$$NGR = (acetate + 2 \times butyrate + valerate)/(propionate + valerate) \tag{6}$$

where VFAs were expressed as a molar proportion, and the appearance of valerate both in the devisor and nominator is due to the fact that valerate may be expected on oxidation to yield 1 mol of acetic acid and 1 mol of propionic acid.

The fermentation efficiency (FE) of energy can be calculated with Equation (7):

$$FE = 0.622 \times acetate + 1.092 \times propionate + 1.56 \times butyrate / (acetate + propionate + 2 \times butyrate) \tag{7}$$

where the VFAs were expressed in molar proportion in total VFA.

### 2.4. Statistical Analysis

The data in this experiment consisted of 2 diet substrates, 6 GABA doses, 5 fermentations per dosage, and three repeated runs. All the statistical analyses used the general linear model procedure of SAS 9.4 with a multiple comparison test (Tukey/Kramer). The model was applied as in Equation (8):

$$Y_{ijk} = \mu + S_i + G_j + (D \times G)_{ij} + B_k + \varepsilon_{ijk} \tag{8}$$

where $Y_{ijk}$ is the dependent variable under examination, $\mu$ presents the overall mean, $S_i$ is the substrate effect where i = 2 for HF and LF, $G_j$ is the fixed GABA dosage level effect for j = 6, $B_k$ is the random effect of repeated batch run (k = 3), and $\varepsilon_{ijk}$ represents the error term. In this paper, least square mean and standard error(S.E.M) were calculated with the least square mean statement to determine linear and quadratic dosage effects of GABA addition. Significance was declared at $p < 0.05$ unless otherwise noted.

## 3. Results

### 3.1. In Vitro Dry Matter Degradability and Kinetic Gas Production

As shown in Table 2, IVDMD was greater in the LF group than the HF group after 72 h in vitro rumen incubation ($p < 0.01$). The GABA addition did alter IVDMD in the LF group, but its addition decreased the IVDMD in the HF group.

**Table 2.** Effect of different addition levels of gamma-aminobutyric acid (GABA) in culture fluids on in vitro dry matter disappearance (IVDMD) and kinetics of gas production of the high-fiber (HF) and the low-fiber (LF) with rumen fluids obtained from dairy cows.

| Item [1] | Diet | GABA Addition (mg) [2] | | | | | | S.E.M | p Value [3] | | | |
|---|---|---|---|---|---|---|---|---|---|---|---|---|
| | | 0 | 10 | 20 | 30 | 40 | 50 | | Diet | L | Q | Diet × GABA |
| IVDMD | LF | 0.88 | 0.87 | 0.89 | 0.88 | 0.87 | 0.89 | 0.004 | ** | ns | ns | ns |
| | HF | 0.85 a | 0.83 ab | 0.81 b | 0.80 b | 0.80 b | 0.81 b | | | | | |
| GP72 (mL/g DM) | LF | 129.8 b | 112.1 c | 131.4 b | 137.6 ab | 150.5 a | 131.5 b | 2.04 | ns | ** | ns | ** |
| | HF | 112.9 | 111.8 | 107.9 | 123.9 | 126.5 | 114.2 | | | | | |
| | | *Kinetic gas production* | | | | | | | | | | |
| A (mL/g DM) | LF | 129.8 ab | 112.1 b | 116.8 b | 129.7 ab | 150.5 a | 125.7 b | 3.18 | ns | ns | 0.01 | ns |
| | HF | 104.0 c | 124.7 b | 109.5 c | 145.3 a | 140.0 a | 135.6 ab | | | | | |
| B | LF | 1.56 ab | 1.68 a | 1.71 a | 1.67 ab | 1.48 b | 1.62 ab | 0.020 | ** | ns | ** | ns |
| | HF | 1.38 ab | 1.43 a | 1.37 ab | 1.36 ab | 1.29 b | 1.41 ab | | | | | |
| C (h) | LF | 6.33 b | 7.13 ab | 7.40 a | 7.34 ab | 7.35 ab | 7.21 ab | 0.115 | 0.02 | ns | 0.04 | ns |
| | HF | 7.01 b | 7.99 a | 7.15 b | 8.09 a | 7.52 ab | 7.36 ab | | | | | |
| RmaxG (mL/h) | LF | 11.20 ab | 10.69 b | 10.35 b | 10.32 b | 12.20 a | 11.40 ab | 0.176 | ns | ns | 0.01 | ns |
| | HF | 11.15 | 10.55 | 11.24 | 11.65 | 11.93 | 11.04 | | | | | |
| TRmaxG (h) | LF | 2.18 b | 3.21 a | 3.29 a | 3.15 a | 2.26 b | 2.73 ab | 0.073 | ** | ns | ** | ns |
| | HF | 1.87 ab | 2.49 a | 2.00 ab | 2.26 a | 1.58 b | 2.16 ab | | | | | |
| RmaxS (/h) | LF | 0.12 ab | 0.13 a | 0.10 c | 0.11 abc | 0.10 c | 0.11 abc | 0.001 | ** | ** | ns | ns |
| | HF | 0.11 a | 0.09 b | 0.10 ab | 0.09 b | 0.10 b | 0.10 b | | | | | |
| TRmaxS (h) | LF | 4.58 bc | 5.29 abc | 5.68 ab | 5.89 a | 4.37 c | 5.04 abc | 0.134 | ** | ns | ** | ns |
| | HF | 3.71 a | 4.24 a | 2.84 b | 3.92 a | 2.90 b | 3.70 a | | | | | |

Means in a row with different superscript letter differ within the same subclass as noted by p value (NS, $p > 0.05$; ** $p < 0.01$). [1] GP72, cumulative gas production at 72 h; (A) asymptotic gas production, (B) sharpness parameter determining the curve shape of the cumulative gas production, (C) the time (h) at which half of A is reached; RmaxG, maximum gas production rate; TRmaxG, the time at which RmaxG is reached; RmaxS, maximum rate of substrate digestion; TRmaxS, the time at which RmaxS is reached. [2] Diluted buffered rumen fluids (75 mL) were incubated for 72 h with 500 mg ground diets as noted in Table 1 with different addition levels of GABA in culture fluids. [3] L, linear effect of GABA addition; Q, quadratic effect of GABA addition; Diet × GABA, the interaction between the diet and GABA addition.

Cumulative gas production profiles are shown in Figure 2. As shown in Table 2, a significant interaction between GABA and diet was observed for GP72; increasing GABA

addition linearly increased GP72 in the LF group ($p < 0.01$), however, the highest $GP_{72}$ occurred in both LF and HF groups with the 40 mg GABA dosage.

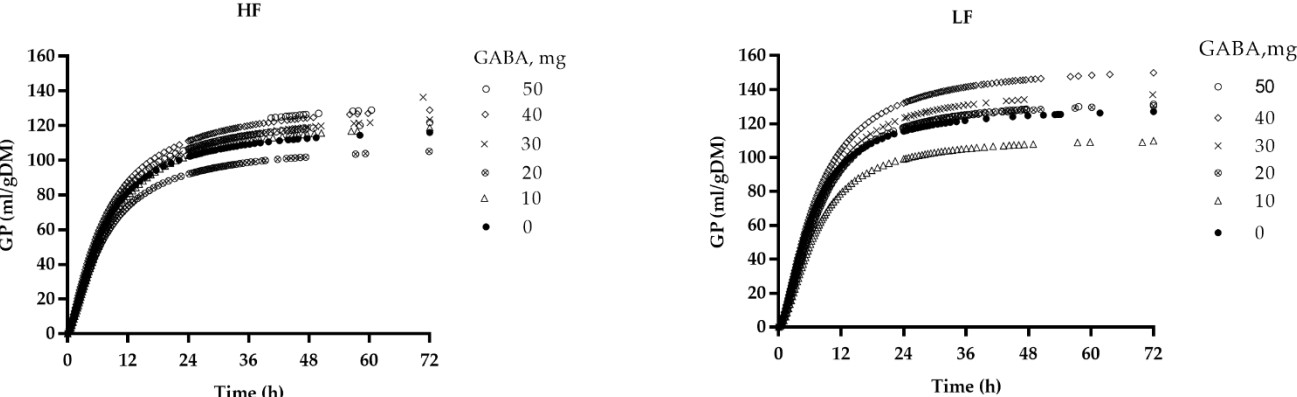

**Figure 2.** Cumulative gas production (GP) profiles of the high-fiber (HF) and low-fiber (LF) diets incubated with rumen fluids with different dosages of gamma-aminobutyric acid (GABA).

Regarding gas production kinetics, increasing GABA addition quadratically increased the asymptotic gas production (A), and the time (h) at which half of A is reached (C) ($p < 0.01$). However, at most dosages, the GABA addition decreased the gas production rate (i.e., RmaxG), as well as the ration digestion rate (RmaxS) to reach maximum fermentation, while the time to reach maximum gas production or substrate digestion increased as GABA addition increased, suggesting that GABA addition could slow down the fermentation process.

### 3.2. Final pH, MCP, and VFA Pattern

Fermentation characteristics of LF and HF diet group after 72 h rumen incubation are shown in Table 3. Increasing GABA addition did alter the final pH and MCP in both the LF and HF diet group. Regardless of GABA addition, the final pH was greater in the HF than the LF group ($p < 0.05$), while MCP was greater in the HF than the LF group ($p < 0.05$).

Increasing GABA addition linearly increased total VFAs ($p < 0.01$) while no statistical difference occurred between the LF and HF groups. A significant interaction occurred between diet and GABA addition; the highest total VFA production in the LF group occurred at 50 mg GABA dosage, while the higher total VFA in the HF group occurred at 30 mg and 40 mg dosages.

Regarding the VFA pattern in molar percentage, acetate, propinate and butyrate were the dominant VFAs in culture fluids for both the LF and HF groups. Increasing GABA addition quadratically increased acetate and butyrate ($p < 0.01$) whereas the interaction effect for acetate occurred between diet and GABA addition ($p < 0.01$). Increasing GABA addition quadratically decreased propionate and the latter was greater in the HF group than in the LF group ($p < 0.01$).

Relatively low molar percentages of valerate and BCVFA (i.e., iso-butyrate and iso-valerate) were also detected in the present study. Regardless of the effect of GABA addition, the LF group in comparison with the HF group presented a greater valerate ($p < 0.01$), while no statistical difference for BCVFA occurred between the LF and HF groups. Increasing GABA addition remarkably decreased valerate in both the LF and HF groups ($p < 0.01$), where a dosage-dependent response was observed. No difference between LF and HF groups was observed for BCVFA. Except for the corresponding GABA- free control, increasing GABA addition increased BCVFA in the LF group ($p < 0.05$), but increasing GABA addition quadratically decreased BCVFA in the HF group, and presented a significant interaction between diet and GABA addition.

**Table 3.** Effect of different addition levels of gamma-aminobutyric acid (GABA) in culture fluids on fermentation characteristics of high-fiber (HF) and low-fiber (LF) diets after 72 h incubation together with rumen fluids obtained from dairy cows.

| Item [1] | Diet | GABA Addition(mg) [2] | | | | | | S.E.M | *p* Value [3] | | | |
|---|---|---|---|---|---|---|---|---|---|---|---|---|
| | | 0 | 10 | 20 | 30 | 40 | 50 | | Diet | L | Q | Diet × GABA |
| Final pH | LF | 6.79 | 6.76 | 6.71 | 6.73 | 6.77 | 6.78 | 0.032 | ** | ns | ns | ns |
| | HF | 6.75 | 6.75 | 6.69 | 6.72 | 6.68 | 6.66 | | | | | |
| MCP (mg/mL) | LF | 1.41 | 1.44 | 1.45 | 1.46 | 1.43 | 1.38 | 0.009 | ** | ns | ns | ns |
| | HF | 1.51 | 1.49 | 1.51 | 1.49 | 1.47 | 1.49 | | | | | |
| Total VFAs (mM) | LF | 89.38 ᶜ | 89.50 ᶜ | 95.36 ᵇᶜ | 98.56 ᵃᵇᶜ | 101.35 ᵃᵇ | 107.19 ᵃ | 2.53 | ns | ** | ns | ns |
| | HF | 90.81 ᵇ | 94.41 ᵃᵇ | 102.23 ᵃ | 101.26 ᵃ | 99.37 ᵃᵇ | 99.37 ᵃᵇ | | | | | |
| | | VFA pattern (%, molar) | | | | | | | | | | |
| Acetate | LF | 68.67 ᶜ | 74.27 ᵃ | 73.99 ᵃ | 70.63 ᵇᶜ | 71.64 ᵇ | 70.81 ᵇᶜ | 0.272 | ns | ns | ** | ** |
| | HF | 71.16 ᵇ | 72.70 ᵃᵇ | 72.21 ᵃ | 73.80 ᵃᵇ | 73.06 ᵃᵇ | 71.71 ᵇ | | | | | |
| Propionate | LF | 13.55 ᵃ | 11.63 ᵇ | 12.53 ᵃᵇ | 12.68 ᵃᵇ | 12.64 ᵃᵇ | 12.78 ᵃᵇ | 0.213 | ** | ns | 0.01 | ns |
| | HF | 13.91 ᵃ | 13.52 ᵃᵇ | 13.30 ᵃᵇ | 13.10 ᵃᵇ | 13.46 ᵃᵇ | 12.74 ᵇ | | | | | |
| Butyrate | LF | 7.60 ᵈᵉ | 7.31 ᵉ | 7.98 ᵈᶜ | 8.19 ᵇᶜ | 8.60 ᵃᵇ | 9.01 ᵃ | 0.065 | ** | ** | ** | 0.01 |
| | HF | 7.59 ᶜ | 7.42 ᶜ | 7.79 ᵇᶜ | 7.39 ᶜ | 8.38 ᵃ | 8.12 ᵃᵇ | | | | | |
| Valerate | LF | 3.86 ᵃ | 1.38 ᶜ | 1.46 ᶜ | 1.43 ᶜ | 1.33 ᶜ | 1.82 ᵇ | 0.039 | ** | ** | ** | ** |
| | HF | 2.35 ᵃ | 1.43 ᵇ | 1.34 ᵇ | 0.81 ᶜ | 1.18 ᵇ | 2.18 ᵃ | | | | | |
| BCVFA | LF | 6.10 ᵃ | 4.54 ᶜ | 4.75 ᶜ | 5.38 ᵇ | 5.83 ᵃᵇ | 5.96 ᵃ | 0.089 | ns | 0.02 | 0.02 | ** |
| | HF | 5.11 ᵇᶜ | 5.23 ᵇᶜ | 5.57 ᵃᵇ | 4.94 ᵇᶜ | 4.73 ᶜ | 4.85 ᶜ | | | | | |
| NGR | LF | 5.30 ᶜ | 6.79 ᵃ | 6.42 ᵃᵇ | 6.24 ᵃᵇ | 6.45 ᵃᵇ | 6.21 ᵇ | 0.074 | ns | ** | ** | 0.03 |
| | HF | 5.44 ᵇ | 5.91 ᵃᵇ | 5.93 ᵃᵇ | 6.44 ᵃ | 6.42 ᵃ | 6.12 ᵃ | | | | | |
| A/P | LF | 5.31 ᵇ | 6.43 ᵃ | 5.49 ᵇ | 5.55 ᵇ | 5.6 ᵇ | 5.55 ᵇ | 0.054 | ** | ns | 0.01 | ** |
| | HF | 5.08 ᵇ | 5.37 ᵃᵇ | 5.41 ᵃᵇ | 5.65 ᵃ | 5.37 ᵃᵇ | 5.58 ᵃ | | | | | |
| FE | LF | 0.712 ᵃ | 0.701 ᵇ | 0.701 ᵇ | 0.706 ᵃᵇ | 0.707 ᵃᵇ | 0.709 ᵃᵇ | 0.0010 | ns | ns | ** | ns |
| | HF | 0.709 | 0.707 | 0.705 | 0.705 | 0.706 | 0.705 | | | | | |

Means in a row with different superscript letter differ within a same subclass as noted by *p* values (NS, *p* > 0.05; ** *p* < 0.01). [1] MCP, microbial crude protein; VFAs, volatile fatty acids; BCVFA, branch-chained fatty acid including iso-butyrate and iso-valerate; A/P, the ratio of acetate to propionate; NGR, ratio of non-glucogenic acids to glucogenic acids; FE, fermentation efficiency. [2] Diluted buffered rumen fluids (75 mL) were incubated for 72 h with 500 mg ground diets as noted in Table 1 with different addition levels of GABA in culture fluids. [3] L, linear effect of GABA addition; Q, quadratic effect of GABA addition; Diet × GABA, the interaction between GABA and Diet.

The overall VFA pattern was estimated with the calculation of the ratio of non-glucogenic to glucogenic acids (NGR). As a result, NGR increased along with the increase in GABA addition, whereas higher GABA addition in the HF group resulted in greater NGR, and lower GABA addition in the LF group presented greater NGR.

### 3.3. Fermentation Gas Composition

Fermentation end-product gas composition profiles during 72 h incubation are shown in Figure 3. The time-course change of fermentation end-products generally showed that carbon dioxide declined from approximately 89% to 74%, and methane increased from approximately 11% to 26%. The statistical responses of gas composition are shown in Table 4.

The fermentation end-product gases including hydrogen ($H_2$), carbon dioxide ($CO_2$), and methane ($CH_4$), were measured in the present study. Since a trace amount of $H_2$ was detected at less than 0.5% in both the LF and HF groups, and did not change significantly, only $CO_2$ and $CH_4$ results are listed in Table 4. In most cases of sampling time, a significant difference in gas composition occurred between the LF and HF groups. As the fermentation proceeded, a step-wise decrease occurred in $CO_2$, while a step-wise increase was observed for $CH_4$ in both the LF and HF groups.

After 72 h incubation, increasing GABA addition significantly and consistently decreased the molar proportion of $CO_2$ in the LF group, and a slight numeric decrease occurred in the HF group, showing an interaction response between diet and GABA addition. Meanwhile, a significant increase in methane proportion was observed in the LF group, and only a numeric increase occurred in the HF group. Regardless of the level of GABA added, methane proportion was consistently greater in the LF than in the HF group, and no interaction was observed between diet and GABA addition.

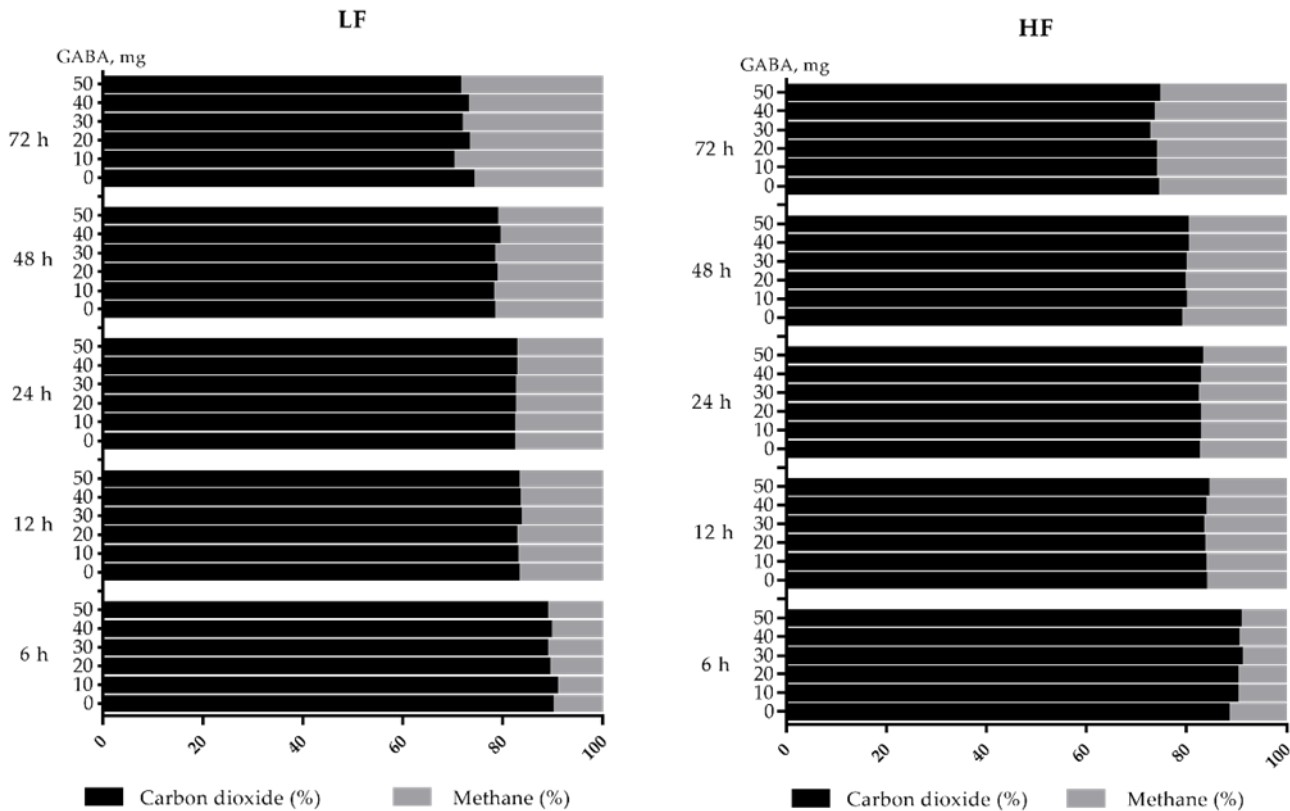

**Figure 3.** Fermentation end-product gas composition profiles of the high-fiber (HF) and a low-fiber (LF) diets incubated with rumen fluids with different dosage of gamma-aminobutyric acid (GABA).

**Table 4.** Effect of different addition levels of gamma-aminobutyric acid (GABA) in culture fluids on fermentation gas composition during 72 h rumen incubation of the high-fiber (HF) and low-fiber (LF) diets incubated with rumen fluids obtained from lactating dairy cows.

| Item | Diet | GABA Addition (mg) [1] | | | | | | S.E.M | *p* Value [2] | | | |
|---|---|---|---|---|---|---|---|---|---|---|---|---|
| | | 0 | 10 | 20 | 30 | 40 | 50 | | Diet | L | Q | Diet × GABA |
| | | CO$_2$ (%) | | | | | | | | | | |
| 6 h | LF | 89.6 [ab] | 90.6 [a] | 89.02 [b] | 88.6 [b] | 89.3 [ab] | 88.6 [b] | 0.33 | 0.03 | ns | ns | ** |
| | HF | 88.1 [b] | 89.9 [a] | 89.8 [a] | 90.7 [a] | 90.1 [a] | 90.6 [a] | | | | | |
| 12 h | LF | 82.9 | 82.7 | 82.5 | 83.3 | 83.1 | 82.9 | 0.22 | ** | ns | ns | ns |
| | HF | 83.6 [ab] | 83.5 [b] | 83.3 [b] | 83.1 [b] | 83.5 [b] | 84.1 [a] | | | | | |
| 24 h | LF | 82.0 | 82.0 | 82.1 | 82.1 | 82.4 | 82.5 | 0.28 | ns | 0.04 | ns | ns |
| | HF | 82.2 | 82.4 | 82.5 | 82.0 | 82.3 | 82.9 | | | | | |
| 48 h | LF | 78.01 | 77.7 | 78.4 | 78.0 | 79.0 | 78.6 | 0.46 | ** | 0.02 | ns | ns |
| | HF | 78.8 | 79.6 | 79.3 | 79.5 | 80.0 | 79.9 | | | | | |
| 72 h | LF | 73.8 [a] | 69.8 [d] | 72.9 [b] | 71.5 [c] | 72.8 [b] | 71.3 [c] | 0.56 | ** | ns | 0.03 | 0.02 |
| | HF | 74.0 | 73.7 | 73.7 | 72.3 | 73.1 | 74.2 | | | | | |
| | | CH$_4$ (%) | | | | | | | | | | |
| 6 h | LF | 10.2 [ab] | 9.2 [b] | 10.7 [a] | 11.1 [a] | 10.4 [ab] | 11.1 [a] | 0.47 | 0.04 | ns | 0.04 | ** |
| | HF | 11.7 [a] | 9.8 [b] | 10.0 [b] | 9.0 [b] | 9.6 [b] | 9.1 [b] | | | | | |
| 12 h | LF | 16.9 | 17.1 | 17.3 | 16.5 | 16.7 | 16.9 | 0.22 | ** | ns | ns | ns |
| | HF | 16.2 [ab] | 16.3 [ab] | 16.6 [a] | 16.7 [a] | 16.3 [a] | 15.7 [b] | | | | | |
| 24 h | LF | 17.9 | 17.9 | 17.8 | 17.7 | 17.4 | 17.4 | 0.28 | ns | 0.04 | ns | ns |
| | HF | 17.7 | 17.4 | 17.4 | 17.9 | 17.6 | 17.0 | | | | | |
| 48 h | LF | 21.9 | 22.1 | 21.5 | 21.8 | 20.9 | 21.2 | 0.47 | ** | 0.02 | ns | ns |
| | HF | 21.0 | 20.3 | 20.6 | 20.4 | 19.9 | 20.0 | | | | | |
| 72 h | LF | 26.1 [d] | 30.1 [a] | 27.0 [c] | 28.4 [b] | 27.1 [c] | 28.6 [b] | 0.55 | ** | ns | 0.03 | ns |
| | HF | 25.9 | 26.2 | 26.2 | 27.6 | 26.8 | 25.6 | | | | | |

Means in a row without common superscript letter differ within a same subclass as noted by *p* values (ns, *p* > 0.05; **, *p* < 0.01). [1] Diluted buffered rumen fluids (75 mL) were incubated for 72 h with 500 mg ground diets as noted in Table 1 with different addition levels of GABA in culture fluids. [2] L, linear effect of GABA addition; Q, quadratic effect of GABA addition; Diet × GABA, the interaction between diet and GABA addition.

## 4. Discussion

### 4.1. The Use of GABA Dosage

The amount of GABA available for absorption in the GI tract of ruminants is difficult to estimate because of the enrichment of GABA in feed, and that broken down and produced by microbes [8]. To interpret our observations accurately, the GABA concentration in food and digestion should be considered without a doubt. The ingestion of GABA by ruminant animals according to the variation of GABA content in feeds has been reported in a number of previous studies. Coenen et al. calculated that GABA ingestion in dairy cows can exceed 50 g per day [24]. Regarding lactating dairy cows consuming 20 kg DM and having an average daily yield of 30 kg milk, we calculated that dietary GABA content could reach up to 2.5 g/kg DM based on the above estimate. Meanwhile, limited data concerning GABA concentrations in the ingesta are available. According to the molar mass of GABA (103.12 g/mol), Wright et al. detected 0.6–8.2 mg/L GABA in bovine ruminal liquid analyzed using thin-layer chromatography [25]. Matsumoto used HPLC to analyze the ruminal liquid of steers and found GABA concentrations ranging from 4.5 to 75 mg/L [9]. Using in vitro incubation systems served with different diets, Thermann et al. reported GABA concentrations ranging between 0.02 and 4.3 mg/L [26]. Assuming a daily fluid flow through the rumen is 240 L for the same dairy cows with 20 kg of daily DM intake and 30 kg of daily milk yield, the estimate of GABA in the rumen could reach up to 15 g per day based on the above digesta levels, indicating a high variability in the values from 15 to 50 g per day.

In the present study, the GABA addition levels were 0 (control group), 10, 20, 30, 40, and 50 mg in a culture fluid with a diet substrate of 500 mg, and theoretical dosage equivalents of GABA were calculated as 20, 40, 60, 80, 100 g/kg diet. Obviously, these doses are indeed 10 times more than 2.5 g/kg DM (50 g per day) based on the above estimates. Considering that GABA can be both broken down and produced by ruminal microbes [26] we should say that the dose levels applied in the present study are mainly for allowing us to detect the possible action effect of GABA on rumen fermentation, and it must be kept in mind that high GABA levels in our study should not be directly recommended for practical use because GABA permeation across the gut wall is actually effective at very low concentrations [27]. According to previous results, our experiment included 500 mg dietary substrate per bottle, which means every bottle may contain 1.25 mg GABA. Or, according to the rumen liquid, the GABA in each bottle may range from 0.015 to 1.875 mg. Because of the high addition of GABA in our study, the content of GABA in the dietary substrate can be neglected. The results in the present study could help us to understand the possible action effects of GABA on rumen fermentation. Of course, we believe further studies on animals through an in vivo study are still needed in the future.

### 4.2. Gas Production and Degradability Responses to Diet and GABA Addition

In vitro fermentation gas production has been widely accepted as a method to detect feed digestion and fermentation kinetics, and various studies have proved this method is both quite convenient and less time consuming compared with in vivo trials. In most studies, IVDMD is usually used as an index to reflect the degradation of the feed in the rumen. According to Dawson and Mayne's study, GABA infusion had no significant effect on DM, modified ADF, and NDF degradability characteristics [13]. Our study also found that there was no significant change in IVDMD with the addition of GABA. But in the HF group, IVDMD was significantly reduced, implying that GABA may impede the fibrolytic activity of microbes in the rumen. Microbes in the rumen tended to utilize the soluble, easily degraded component, but dietary fiber composed from structural carbohydrate will to some extent slow down the rumen fermentation process [28]. For this reason, the HF group showed a lower IVDMD and $GP_{72}$ compared with the LF group in the present study.

In vitro fermentation gas production has been used to determine feed digestion characteristics in recent years [18]. In the present study, $GP_{72}$ significantly increased with GABA addition in the LF group. In a previous study, Reyes-García et al. revealed that GABA could

promote the fungal growth of *Candida albicans* which may explain the increase in $GP_{72}$ [29], and this could allow us to speculate that the GABA addition could also boost the growth of fungi or other bacteria in the rumen and enhance some metabolic processes of microbes. From these results, the fiber content in diet and the addition of GABA presented an interaction effect on gas production: increasing GABA addition resulted in a significant increase in $GP_{72}$ for the LF group but not for the HF group. The authors also observed a decrease in IVDMD in the HF group in response to GABA addition, implying that GABA addition might stimulate microbes which specialize in non-structural carbohydrate degradation.

With the addition of GABA, a significant change in the kinetics of gas production was noticed. Both RmaxG and RmaxS reflect the time taken for digestion rate to reach maximum level [17]. In the present study, the asymptotic gas production in both the LF and HF groups were not decreased by GABA addition, implying that GABA addition had no detrimental effect on extent of nutrient fermentation. The lowered RmaxG or RmaxS in response to GABA implicated that the GABA addition could somewhat slowdown speed of the fermentation process, and such a deceleration obviously had a benefical action on rumen fermentation especially when high-starch or low-fiber diet fed to ruminants, and put animal at risk of rumen acidosis.

### 4.3. In Vitro Rumen Fermentation Characteristic Responses to Diet and GABA Addition

Stable rumen pH at 6.6-6.8 provided a suitable environment for rumen microorganisms. In the present study, rumen pH was maintained between 6.71 and 6.76 not only in the HF group both also in the LF group regardless of the level of GABA addition. This result was consistent with Dawson's study, in which the GABA infusion did not change rumen pH compared with the control group [30], and this could be explained by the lowered RmaxG or RmaxS in response to GABA as noted in above Section 4.2.

In the rumen, the MCP level usually reflects the growth rate and the population of rumen microbes [17,31]. The synthesis of MCP is affected by the content of protein, nonstructural carbohydrates, and structural carbohydrates in the diet [32]. Thus, the degradation of protein could modulate the flow of MCP [33]. Usually, an increase in the ratio of nonstructural carbohydrate to NDF would increase the synthesis of MCP. In the present study, MCP showed no significant change with the addition of GABA, suggesting that GABA did not inhibit the growth of rumen microbes, though GABA is known as the most abundant inhibitory neurotransmitter in the mammalian brain [1,2].

VFAs are the end products in rumen fermentation, and mainly include acetate, propionate butyrate, valerate, and branch-chained fatty acids (BCVFAs, including iso-butyrate and iso-valerate), and they supply the main metabolizable energy. The production and the proportions of VFAs can somewhat reflect the metabolism status of rumen microorganisms. In our results, total volatile fatty acids linearly increased with GABA addition, suggesting the degradation of carbohydrates in the rumen and energy-use efficiency were promoted [34]. However, this result was not consistent with Dawson's results [13]; they noted that the infusion of GABA (2, 4, 6 g/kg DM) in the rumen did not cause any significant change to total VFA [30]. Such inconsistent results could be explained by the fact that the production and absorption of VFA was actually a dynamic process in Dawson's in vivo study when compared with the present in vitro study. The increase in VFAs in response to GABA addition could not be detected in vivo as much as they can in vitro. In the present in vitro study, the absorption of VFA can be neglected, and can give a more precise description of rumen fermentation parameters.

Acetate, propionate and butyrate are the three major VFAs in the rumen which offer up to 80% of their maintenance energy [35]. GABA addition in both the LF (+8.15%) and HF (+3.71%) groups increased the proportion of acetate in total VFAs, and such results were consistent with Dawson's study [13,30]. Meanwhile, the ratio of acetate to propionate (A/P) increased in the LF group to up to 21.09%, and up to 11.00% in HF group. As revealed in previous articles, acetic acid produced by the rumen fermentation of carbohydrates is the main precursor of synthesis of fatty acids in the mammary gland [36]. The improvement

of the efficiency of milk fat synthesis could lead to more milk fat and improve the milk quantity. What's more, the increase in acetic acid, butyric acid, and the ratio of acetate to propionate as well as NGR implied that GABA addition could shift the pattern of carbohydrate fermentation in the rumen towards more non-glucogenic acid production (i.e., acetate and butyrate). As end products of dietary protein breakdown, iso-butyrate and iso-valerate are derived via transamination from valine and leucine, respectively. The cofactor of branched chain volatile fatty acids (BCVFA) is used by most rumen microorganisms. Cellulolytic microorganisms primarily utilize BCVFA as the main source of carbon chains for growth [37]. Usually, the increase in amount of protein in diets could increase the concentration of BCVFA. Thus, in the present study, the LF group presented higher BCVFA than the HF group. The lowered BCVFA in the present study implied that the GABA addition could not alter protein metabolism in term of amino acid transamination.

Unlike the present study, which focuses on the effect of GABA in the rumen, a previous study with rumen-protected GABA noted that dietary addition of rumen-protected GABA did not change the content of milk protein and fat in Chinese Holstein lactating cows [10]. Although GABA acts as an interspecies signal in host–microbe interactions, there is still no clear explanation of how the structural characteristics of selective eukaryotic GABA receptors and bacterial GABA sensors may cause this [38]. Whether or not such a fermentation shift as observed in the present study could happen in vivo and increase milk fat in milk is yet to be justified.

### 4.4. Fermentation Gas Composition Responses to Diet and GABA Addition

The rumen is open to the external environment, which forms a continuous flow of material into and out of the rumen [19]. Methanogenic archaea, considered to be the earliest organisms originating on our planet, are phylogenetically widespread and represent three of the seven classes in the phylum Euryarchaeota. These rumen microorganisms utilize the $H_2$ and $CO_2$ produced by the protozoa, fungi, and bacteria from the catabolism of hexoses to produce $CH_4$ and generate ATP [39], which benefits the donors by providing an electron sink for reducing equivalents to minimize the partial pressure of $H_2$ inside the rumen. Mackie et al. noted that gas composition in the rumen mainly consisted of 65% $CO_2$, 27% $CH_4$, 7% $N_2$, 0.6% $O_2$ and 0.2% $H_2$ [40]. In the present in vitro study, we did not detect any $N_2$ or $O_2$. The proportion of $H_2$ was not affected by the diet substrate and the GABA. Thus, the proportion of $H_2$ was not listed in the present study. As fermentation proceeded in the present study, the step-wise decrease in $CO_2$ and the corresponding step-wise increase in $CH_4$ in both the LF and HF groups implied that methanogens would not be very active at the beginning of the fermentation process, but would be quite active during subsequent fermentation to minimize the partial pressure of $H_2$ inside the rumen. This could also be the reason why a trace amount of $H_2$ was difficult to detect. The results that $CH_4$ was consistently greater in the group LF than the HF group implied that the establishment and maintenance of a stable population of methanogens might be mainly affected by the type of diet. Since the enteric fermentation of ruminant animals is blamed for global warming due to the production of both $CH_4$ and $CO_2$ [41], the significant increase in $CH_4$ in response to GABA addition, especially in the LF group, would not be welcomed from a practical viewpoint of animal production although its relative action mode was not clear based on the limited information in the present study.

### 5. Conclusions

Increasing GABA addition decreased dry matter digestibility in the high-forage diet, but did not affect that of the low-forage diet. Kinetic gas production analysis implied that increasing GABA addition could slow down the speed of the fermentation process. Regardless of differences in forage-to-concentrate ratio, the addition of GABA did not alter pH, but did enhance rumen VFA production and shifted the rumen fermentation pattern towards more acetate and butyrate and less propionate production. Additionally, increasing GABA addition increased methane production, and such an effect was more

pronounced in low-forage than in high-forage diets. Whether or not the above in vitro responses to GABA addition could happen in the rumen will need to be justified with further in vivo study.

**Author Contributions:** Conceptualization, Y.-L.W. and H.-J.Y.; methodology Z.-H.Z.; software, Y.-L.W. and Z.-H.Z.; validation, F.Z. and W.-J.L.; formal analysis, Y.-L.W. and Z.-J.C.; investigation, Y.-L.W. and W.-K.W.; resources, S.-L.L. and Q.-C.W.; data curation, Y.-L.W., W.W. and H.-J.Y.; writing—original draft preparation, Y.-L.W. and H.-J.Y.; writing—review and editing, Y.-L.W., W.-K.W. and H.-J.Y.; visualization, H.-J.Y.; supervision, H.-J.Y.; project administration, H.-J.Y.; funding acquisition, H.-J.Y. All authors have read and agreed to the published version of the manuscript.

**Funding:** The authors acknowledge financial support from the National Natural Science Foundation of China (grant no. 31572432).

**Institutional Review Board Statement:** The study was conducted according to the guidelines of the Declaration of Helsinki, and approved by the Institutional Review Board of Institutional Animal Care Committee of China Agricultural University (protocol code CAU20171014-1 and date of approval 25 September 2020).

**Informed Consent Statement:** Not applicable.

**Data Availability Statement:** Data is contained within the article.

**Conflicts of Interest:** The authors declare no conflict of interest.

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
