# Peer review of "The Effect of γ-Aminobutyric Acid Addition on In Vitro Ruminal Fermentation Characteristics and Methane Production of Diets Differing in Forage-to-Concentrate Ratio"

_fermentation, doi:10.3390/fermentation9020105_

Round 1

Reviewer 1 Report

I suggest that the conclusion in the summary… These results implicated that dietary addition of GABA may promote milk fat production in lactating cows. ..  It is not supported, since in this investigation milk production or its composition was not measured, and the addition of GABA may or may not improve it.

The object of the present study was to attempt to elucidate the effects of GABA in different fibre content diet on in vitro rumen digestion, rumen fermentation characteristic  and the gas compositions and, therefore, it must be concluded based on the objective.

The experimental diets, LF and HF contained very different CP concentrations (116.9 and 175.8 g/kg DM, respectively). Branched-chain AGVs are known to come from the fermentation of amino acids. However, you do not make any reference or discussion to the effect that this difference could have on the effect of GABA on rumen fermentation.

I suggest that you have a little discussion about whether or not the CP level had any effect on rumen fermentation when GABA was added.

I agree with you that…..we should say that the dose levels applied in the present study are mainly for allowing us to detect the possible action effect of GABA on rumen fermentation, and it must be kept in mind that high GABA levels in our study should not be directly recommended for practical …and this is one of the reasons why I do not agree with the conclusion that GABA may promote milk fat production in lactating cows.

You mention that...In the rumen, MCP represented a major source of amino acids to ruminants which can indirectly reflect the growth rate and the population of rumen microbes,… but you do not mention if the level of protein in the diets affected in any way the results that you discuss with the addition of GABA. I suggest a brief explanation about it

In the conclusions I kindly suggest you consider the following wording. The addition of GABA enhanced rumen VFA production and change the rumen-fermentation pattern exhibited a shift to produce more acetate. The efficiency of GABA depended on the content of fibre in diets. In the present study, the GABA addition increased the gas production by 15.9% in low-fibre diet which indicated the greater utilization of substrates. Thus, whether the GABA addition applied as a feed additive in dairy industry can be practicable so as to improve economic benefits is  required to set up a cost-effective dose in practice.

My suggestion is based on the fact that milk production or composition was not determined and that the addition of GABA may or may not improve milk fat simply because it has registered a higher acetic acid content in in vitro tests.

Author Response

Dear Reviewer,

We appreciate you for your precious time in reviewing our paper and providing us valuable comments. It was your valuable and insightful comments that led to possible improvements in the current version. We has carefully considered the comments and tried our best to address every one of them.

Yanlu

Reviewer 2 Report

My only concern about the MS is that authors performed only 1 period with 5 observations per treatment, which the ideal for in vitro assays would be to have performed at least 3 incubations periods. Therefore, conclusion about your results should be interpreted with caution!

Author Response

Dear reviewer,

We appreciate you for your precious time in reviewing our paper and providing valuable comments. It was your valuable and insightful comments that led to possible improvements in the current version. We have carefully considered the comments and tried our best to address every one of them..

Yanlu

Reviewer 3 Report

The manuscript needs extensive English revision to clarify the whole paper. It is however a well designed study and interesting aspects of the fermentation are taken into consideration. I suggest you also revise the sections where you easily link the in vitro results with the animal. Make sure you underline that the in vivo results could be different due to other factors in play. Below I suggest major revision but mainly for the editing of the English and not the work itself.

Few line comments below:

Line 45: has

Line 47-50 revision of English needed

Line 50: was that the only “controversial” study

Line 52: objective?

Line 52: clarify and justify why you now focus on fiber and in vitro, compared to the previous studies and results that you mention. If studies in vitro were done you need to cite those

Line 57: need to introduce the set up of the study before describing the HF and LF ration. You also need to justify why you chose these specific diets

Line 59: better to indicate %

Line 63: check English

Line 71: check English

Line 73: need to justify these levels somewhere in the text… where do they come from?

Line 71-74: check English

Line 73: what is the final concentration and/or the amount per g/DM substrate?

Line 78: I suggest you add a table about this and remove it from the text. You also will need to specify the vitamin mineral mix composition in a footnote.

Line 269-270: expand on this conclusion please

Line 304: there is no point in talking about rumen protected GABA here and elsewhere in the paper

Author Response

Dear Reviewer,

First of all, great thanks for your time to give us review comments. Following your nice suggestion, we have check the whole manuscript point by point, and the revised places have been highlighted . Hopefully the revised version could be acceptable. All the corrections have been highlighted in yellow.

Yanlu

Reviewer 4 Report

see the attachment

Author Response

Dear Reviewer,

We appreciate you for your precious time in reviewing our paper and providing valuable comments. It was your valuable and insightful comments that led to possible improvements in the current version. We has carefully considered the comments and tried our best to address every one of them.
